# NADES Compounds Identified in *Hypoxis hemerocallidea* Corms during Dormancy

**DOI:** 10.3390/plants11182387

**Published:** 2022-09-13

**Authors:** Motiki M. Mofokeng, Gerhard Prinsloo, Hintsa T. Araya, Stephen O. Amoo, Christian P. du Plooy, Phatu W. Mashela

**Affiliations:** 1Agricultural Research Council—Vegetable, Industrial and Medicinal Plants (ARC-VIMP), Private Bag X293, Pretoria 0001, South Africa; 2Green Technologies Research Centre, University of Limpopo, Private Bag X1106, Sovenga 0727, South Africa; 3Department of Agriculture and Animal Health, University of South Africa, Private Bag X6, Johannesburg 1710, South Africa; 4Department of Botany and Plant Biotechnology, University of Johannesburg, P.O. Box 524, Auckland Park, Johannesburg 2006, South Africa

**Keywords:** propagation, choline, lactose, organic acids, succinate, propylene glycol, sugars, NADES

## Abstract

Soaking *Hypoxis hemerocallidea* corms in distilled water improved the propagation and development of cormlets, suggesting the potential leaching-out of inhibitory chemical compounds. To investigate the presence of inhibitory compounds, nuclear magnetic resonance (NMR) spectral data of the leachate from dormant *H. hemerocallidea* corms were obtained using a 600 MHz ^1^H-NMR spectrometer. The ^1^H-NMR analysis led to the identification of choline, succinate, propylene glycol, and lactose, as inhibitory compounds. These four chemical compounds are part of the “Natural Deep Eutectic Solvents” (NADES) that protect plant cells during stress periods, each of which has the potential to inhibit bud growth and development. These compounds are supposedly leached out of the corms during the first rain under natural conditions, possibly accompanied by changes in the ratios of dormancy-breaking phytohormones and inhibitory compounds, to release bud dormancy. The identified chemical compounds heralded a novel frontier in the vegetative propagation of *H. hemerocallidea* as a medicinal plant, and for its enhanced sustainable uses.

## 1. Introduction

Plant growth regulators (PGRs) play critical roles in plant growth and development; however, certain endogenous chemicals interfere with the synthesis, transport, or action of PGRs, and they are technically referred to as growth inhibitors or retardants [1]. Plant growth regulators such as abscisic acid (ABA) induce and maintain dormancy in buds and seeds of various plant species, whereas ethylene has antagonistic effects on the growth-promoting gibberellic acid (GA_3_) [2]. Similarly, certain sugars have the ability to prevent the perception of GA_3_, in what had since been referred to as sugar-PGR ‘cross-talking’, thus serving as growth inhibitors [2]. Lactose, for example, had inhibitory effects on the micropropagation of turmeric [3]. During dormancy, underground organs of most plants undergo metabolic changes such as altered sugar concentrations, and the accumulation of free amino acids, compatible solutes, and other chemical compounds [4]. Compatible solutes, which are also called osmoprotectants, are synthesised during dehydration to enhance plant stress tolerance [5]. For example, chlorocholine chloride (CCC) inhibits growth [6], whereas choline induces stress tolerance in most plants [7]. Chlorocholine chloride was also reported to have a protective effect on stevia plants grown under drought stress, by inhibiting gibberellin biosynthesis [8]. Succinate, which accumulates in various organs when plants are exposed to water stress [9], could play a role in dormancy sustenance. Drought tolerance in plant species, as induced in plants starting to experience winter dormancy, correlates with the induction of sugars and is generally believed to be involved in the formation of “Natural Deep Eutectic Solvents” (NADES) such as cholines, prolines, and other organic acids [10]. The NADES are special mixtures of solids that remain in a liquid state at and below ambient temperatures, with physiological significance in plants due to their unique properties of solubilising other metabolites and bioactive constituents [11]. Natural compounds synthesised in plants can have inhibitory or stimulatory effects on various plant species. Aqueous extracts of peppermint limited seed germination of select vegetable and cereal seeds, except for *Phaseolus vulgaris* and *Zea mays* seeds, in which the extracts had a stimulatory effect [12]. Palacios et al. [13], screened 71 plant extracts for their inhibitory effects on common oats and wild radish, with some of the extracts exhibiting from 80 to 100% inhibition on germination.

*Hypoxis hemerocallidea* Fisch. & C.A. Mey. (Hypoxidaceae) is one of the indigenous South African medicinal plants that has generated interest for product development, both locally and internationally and, therefore, is one of the heritage plant species [14]. The corm of *H. hemerocallidea* enables the plant to survive cold conditions and drought. Under such conditions, leaf senescence and plant dormancy are observed [15]. Some phytochemicals from *H. hemerocallidea* corms including daucosterols, beta-sitosterol, and hypoxide have been associated with the therapeutic activities of this plant [16]. *Hypoxis hemerocallidea* is a common ingredient of popular herbal remedies consumed by people living with HIV/AIDS in South Africa as an immune booster [17,18,19]. The plant is also used for the potential treatment of bronchial asthma [20], cancer [21,22], and certain sexually transmitted diseases [23,24]. In the light of its current high demand coupled with its unsustainable methods of harvesting from the wild, the conservation of this plant has become very important for ensuring its survival and sustainable use. Affordable propagation methods, such as chipping and scooping to remove the correlative inhibition from the apical buds [25], were developed and found to be successful for the development of cormlets [26].

Soaking *H. hemerocallidea* “mother corms” in distilled water for 120 min improved the development of cormlets [27], suggesting the potential leaching of inhibitory chemical compounds. The endogenous PGR content and their balance in plants are essential for achieving desired morphological responses, such as root and shoot regeneration, along with subsequent growth and development [25]. Generally, *H. hemerocallidea* corms are dormant in winter and sprout as soon as they receive rain. The breaking of corm dormancy resulting in corm sprouting could be due to a leaching effect of the rain on corm-endogenous inhibitory compounds. The nature of such inhibitory chemicals in *H. hemerocallidea* corms had not been investigated. The objective of this study, therefore, was to investigate the presence of, and to identify, leachable inhibitory chemical compounds in dormant *H. hemerocallidea* “mother corms”.

## 2. Results

The ^1^H-NMR spectrum (Figure 1) illustrated the metabolite profile of the leachate from *H. hemerocallidea* corms.

Four chemical compounds annotated from the leachate with potential inhibitory or growth-retarding effects were numbered 1, 2, 3, and 4, with 1 and 2 representing organic acids, whereas 3 represented peaks of the sugar region, and 4, the peaks for propylene glycol (Figure 2).

The annotation of the identified chemical compounds, with the published NMR chemical shifts, and the Chenomx and human metabolome database spectral regions or chemical shifts are summarised in Table 1. Chemical compound **1**, annotated as choline, had a chemical shift at 3.192 ppm. Chemical compound **2**, annotated as succinate, had a chemical shift at 2.42 ppm, and compound **3**, annotated as lactose, showed peaks in the sugar region at 3.28, 3.55, 3.75, 3.85, 3.95 and the anomeric proton at 5.12 ppm. Compound **4**, annotated as propylene glycol had a chemical shift for a doublet at 1.14 and 1.16 ppm, and other peaks at 3.4 and 3.5 ppm. The spiking of the leachate with the pure standards of propylene glycol (Appendix A), succinate (Appendix A), choline (Appendix A), lactose (Appendix A), aligned with identified peaks. The presence of gallic acid could not be confirmed as the peaks of the leachate sample and the pure standard of gallic acid did not align (Appendix A).

Using 3-(trimethylsilyl)propionic acid sodium salt (TSP) at 0.1% as a reference compound, the compounds were quantified using the signature peaks for each compound. Lactose was found to be high in concentration and propylene glycol was the lowest (Table 2).

The significant inhibitory effect of the *H. hemerocallidea* leachate was demonstrated through tomato seed germination. A 23% reduction in the final germination percentage was recorded in leachate-treated seeds in comparison to the control (seeds germinated with distilled water) (Figure 3). The seeds exposed to the leachate also showed comparatively slow germination, as indicated by the significantly increased mean germination time recorded. The lower the mean germination time, the faster the germination. Moreover, the germination index, coefficient of velocity of germination, and germination rate index were all significantly reduced by the leachate treatment.

## 3. Discussion

Analysis of the leachate from *H. hemerocallidea* corms submerged in distilled water led to the identification of four main chemical compounds, namely, choline, succinate, propylene glycol, and lactose, each with the potential to inhibit bud growth and development [6,8,30]. Choline has growth-retarding effects on various plants and generally inhibits the synthesis of GA_3_, renowned for its dormancy-breaking effects [31]. Chlorocholine-chloride (CCC) and acetylcholine, both having similar chemical structures to that of choline, have strong inhibitory effects on the growth of many plants, although their content is influenced by environmental conditions and plant developmental phases [6]. Choline was reported to be a neurotransmitter as a component of acetylcholine and its role is facilitated by a set of enzymes and receptors linked to acetylcholine as a neuronal mediator involved in plant physiological processes [6]. Chlorocholine chloride inhibited the (−)-kaurene synthesis, an intermediate in GA_3_ synthesis, thus inhibiting plant growth and development [32,33,34]. The effect of acetylcholine has been linked to its regulation of ionic fluxes [35] by intervening with the functions of potassium cations (K^+^), thus affecting the root to shoot growth [36]. For example, corms of *Gladiolus tristis* L. exposed to 1% KNO_3_ sprouted in 23 days compared to 45 days it took in the control treatment [37], a possible indication that potassium could be critical in inducing corm bud development and that choline’s effect on the K^+^/Na^+^ ratio could induce bud dormancy in the corms of the test plant. Generally, CCC had inhibitory effects on the sprouting of saffron corms compared with the bud breaking effects of GA_3_ [30].

Cutting the *H. hemerocallidea* corms to expose the reproductive buds and soaking in water, improved the development of cormlets, to a similar extent as soaking in GA_3_ [26]. Since *H. hemerocallidea* plants are dormant in winter, choline could act as a neurotransmitter in the corms by sending signals to induce underground bud dormancy under changing environmental conditions, as it is highly concentrated in meristematic and differentiating tissue [38]. For example, through its association with phytochrome-mediated processes, choline could be involved in the circadian signalling network, which is sensitive to the photoperiod and light quality [39] or perceiving dormancy-inducing short days [40], and thus, inducing dormancy in the test plant. The onset of short days could have induced growth cessation and transition to dormancy [41] by downregulating GA_3_ biosynthesis through the inhibition of (−)-kaurene synthesis and upregulating the functioning of repressor genes of meristematic activity [40].

Succinate is a compound that is oxidized to fumarate in the mitochondrion during respiration [42,43]. A decrease in plant-cell respiration is linked to the reduced activity of succinate dehydrogenase (SDH), an enzyme that oxidizes succinate, thus, increasing concentrations of succinate in the mitochondrion and decreasing energy production [42]. Modified stems of most plants experience substantial metabolic changes when entering dormancy, such as decreased respiration [4,44], whereas respiration and SDH are upregulated during dormancy release [45]. The interruption of SDH activity induces signalling patterns such as high-intensity reactive oxygen species (ROS) production related to the growth and development or stress response in plant tissues [46,47,48,49,50]. The partial inhibition of SDH decreased shoot and root growth in Arabidopsis and completely blocked hypocotyl elongation and seedling establishment [42]. Succinate dehydrogenase activity in potato tubers declined during tuber maturation just before dormancy and increased during tuber initiation after dormancy release [51,52]. Succinate dehydrogenase and α-ketoglutarate dehydrogenase (KDH) mediate the regulatory action of acetylcholine in the mitochondria through parasympathetic nervous system regulation and oxidation in the mitochondria, thus, forming part of the hormonal system [53], possibly through the ‘cholinergic system’. Succinate was among the chemical compounds that increased significantly when benthi (*Nicotiana benthamiana* Domin) plants were under severe water stress compared to those under moderate water stress [8]. Water stress in *H. hemerocallidea* corms during early winter resulted in increased succinate content, thereby sustaining the dormancy state. Succinate dehydrogenase activity could have been reduced as the plant entered dormancy due to changes in photoperiod [54], thus the accumulation of succinate, which could have played a role in the signalling system. Due to the reduction in respiration rate, energy production is reduced and, thus, no growth or development was observed among the *H. hemerocallidea* plants.

Lactose is a specific product of the mammary gland, the detection of which has been controversial since 1949 [55]. The biosynthesis and pathway of sugar accumulation, especially lactose, during plant stress are enigmatic [56]. β-galactosidase, also called lactase, is an enzyme of microbial, plant, and animal origin that breaks down lactose into two monosaccharides, galactose and glucose [57]. In plants, β-galactosidase is mainly found in almonds, peaches, apples, and apricots, in which it plays a critical role in plant growth [57]. Changes in the activity of plant enzymes that metabolise sugars can induce accumulation in plant cells, which can be reversed when the stress factor is removed [56]. Sugars are important in metabolism-related signalling mechanisms that regulate the growth activity and dormancy cycle [58]. For example, when temperatures are low, an imbalance in energy flow from the source to the sink is created, affecting the biochemical reactions and carbohydrate metabolism in plant cells [55] through ‘rate-limiting steps’ [59]. To be fully active, β-galactosidase requires K^+^ or Na^+^ [60]. The increased accumulation of lactose in the test plant could have been due to the reduced activity of β-galactosidase, resulting in lower K^+^ in the plant cells and leading to reduced growth activity and the induction of the energy-release ‘rate-limiting step’. Sugars protect plant cells during dehydration by “glass formation”, where, instead of plant solutes forming crystals, the presence of sugars results in their formation of supersaturated liquids with mechanical properties of solids [61]. “Glass formation” under drought conditions could be as a result of the formation of NADES consisting of sugars, choline, and other organic acids, and it is accompanied by a decrease in chemical reactions [10]. Choline, lactose, and succinate are among other plant metabolites identified as NADES compounds [11].

Propylene glycol is an aliphatic alcohol used as a deep eutectic solvent (DES) [62,63]. Other than its use as a DES, it has applications in the cosmetic industry as a skin-conditioning agent (humectant), viscosity-decreasing agent, solvent or fragrance ingredient; as a food additive; in pharmaceutical products; electronic cigarette liquids, and manufacturing [64,65]. There is not much work reported on propylene glycol in plants and its effect on plant dormancy, which may need further investigation. Propylene glycol toxicity has been reported previously [66,67], and investigating non-toxic substitutes has been recommended [68]. Propylene glycol is usually a petroleum-based chemical, and efforts have been made to produce it from soybean and canola as an environmentally friendly alternative [69].

Tomato is one of the most commonly grown fruit vegetables by commercial, subsistence, resource-poor farmers and home gardeners in South Africa. The majority of the population consumes it in diverse ways [70]. Tomato was employed to determine the potential inhibitory effect of *H*. *hemerocallidea* leachate on plant growth. The leachate inhibited all the germination parameters/indicators recorded, in comparison to the distilled water-treated seeds. However, the extract needs to be tested on more plant seeds and on releasing dormant buds as its effect may vary between different plant species. Biologically active substances produced by plants may have selective effects on other species; they may have a complete inhibitory effect or slow [12] growth slightly. For example, out of 71 tested extracts, only *Baccharis salicifolia*, *Ophryosporus charua*, and *Angelphytum aspilioides* extracts inhibited germination of common oats and wild radish [13].

The identified NADES compounds may also be of significance for the plant’s medicinal uses. The presence of compounds such as succinate could explain some of the uses of *H. hemerocallidea* corms in traditional medicine as an immune booster. For example, in animals, succinate modulates blood pressure and immune- and blood-cell function due to its role in blood and immune signalling prerequisites for platelet aggregation [43,71]. Succinate also plays a beneficial role in the coronary artery and heart diseases and related reperfusion injuries [72]. However, succinate accumulation in the body can lead to hypertension, obesity, and liver damage [71], which could explain the reported risk of kidney damage from the prolonged oral intake of *H. hemerocallidea* extracts [73]. Lactose intolerance has been reported to be a problem for many individuals who cannot hydrolyse lactose in the small intestine, resulting in gut symptoms such as diarrhoea, constipation, vomiting, nausea, and others [74]. Choline, on the other hand, was found to be effective in memory enhancement and resistance to cognitive decline [75] as well as in inhibiting airway inflammation [76], indicating a new potential use of *H. hemerocallidea* in the treatment of asthma and neurodegenerative diseases. The production of propylene glycol from plant materials could decrease greenhouse gasses by approximately 60% [69], and it may also provide solutions to the toxicity of petroleum-based propylene glycol, which could be an advantage to the health industry.

## 4. Materials and Methods

### 4.1. Plant Material and Leachate Preparation

Mature *H. hemerocallidea* corms (accession number = M2010/013) were lifted from the soil in the medicinal plant genebank of the Agricultural Research Council in Roodeplaat, Pretoria, South Africa. Corms (weighing 250 g on average) were lifted in late winter (August), rinsed in distilled water to remove soil particles, blotted dry using laboratory paper towels, and further dried overnight by leaving the corms on laboratory benches. Thereafter, each corm was cut into four equal pieces (Figure 4), using a clean, sharp knife and the pieces were submerged in 250 mL distilled water in a soaking bowl for 16 h, by which the distilled water had changed colour from a clear to dark brown/blackish liquid. The corms were then removed from the soaking bowl, and the dark brown/blackish solution was collected for analysis.

### 4.2. Reagents

Deuterated methanol, potassium dihydrogen phosphate, and 3-(trimethylsilyl)propionic acid sodium salt (TSP) were obtained from Merck Life Science (Pty) Ltd. (Modderfontein, South Africa). Pure standards of succinate (≥98% purity), lactose (≥98% purity), propylene glycol (99% purity), gallic acid (≥99% purity), and choline (≥99% purity) were also obtained from Merck Life Science (Pty) Ltd. (Modderfontein, South Africa).

### 4.3. Leachate Analysis and Compound Identification

The 250 mL leachate was dried in a SpeedVac^TM^ vacuum concentrator (ThermoFisher Scientific, Waltham, MA, USA) for 24 h, with a 50 mg dried sample weighed out into 2 mL Eppendorf tubes for extraction and analysis. Thereafter, 0.75 mL methanol-d_4_ (CD_3_OD) and potassium dihydrogen phosphate (KH_2_PO_4_), buffered in deuterium water (D_2_O) (pH 6.0), containing 0.1% (*w*/*w*) 3-(trimethylsilyl)propionic acid sodium salt (TSP), were added to samples. The mixtures were vortexed at room temperature for 1 min, ultra-sonicated for 20 min, and then centrifuged for another 20 min at 10,000 rpm. The supernatant from each tube was transferred to a 5 mm nuclear magnetic resonance (NMR) tube (Norell, Sigma-Aldrich, St. Louis, MO, USA) for analysis. The NMR spectral data were obtained using a 600 MHz ^1^H-NMR spectrometer (Varian Inc., Santa Clara, CA, USA), with 32 scans recorded. The human metabolome [77], Chenomx, and other references were used for the annotation of compounds present in the leachate.

To confirm the annotated compounds, samples were spiked with pure standards of succinate, lactose, propylene glycol, gallic acid, and choline with subsequent NMR analysis. For this analysis, 100 mg of leachate was extracted with 0.75 mL methanol-d_4_ (CD_3_OD) and KH_2_PO_4_, buffered in D_2_O (pH 6.0), containing 0.1% (*w*/*w*) TSP. This was repeated to yield five NMR tubes (one control and four tubes for each of the standard compounds). In each tube, 500 µL of the extract was added. Different standards of lactose (0.684 mg/mL), choline (0.1 mg/mL), succinate (190.240 mg/mL), gallic acid (0.018 mg/mL), and propylene glycol (0.2 mL/mL) were prepared in 0.75 mL methanol-d_4_ (CD_3_OD) and KH_2_PO_4_, buffered in D_2_O (pH 6.0), containing 0.1% (*w*/*w*) TSP, and 500 µL of each standard solution was added to one of the NMR tubes containing the leachate extract. The final volume per NMR tube was, therefore, 1 mL, except for the control sample that only contained 500 µL of the leachate extract. An increase in the heights of the peaks of the annotated compounds was considered as confirmation of the presence of the compounds in the leachate at identification level 1 [78].

### 4.4. Compound Quantification

The compounds identified in the study were quantified using the software Chenomx (Chenomx, Edmonton, AB, Canada). This software matches compounds to comprehensive spectral reference libraries to identify compounds but also to accurately measure concentrations of compounds in an NMR sample. TSP (0.1%) was used as reference standard in the quantification of succinate, propylene glycol, lactose, and choline. For each compound, signature peaks were identified as those that did not overlap with other peaks and were positively identified as peaks belonging to the compounds after the spiking of the compounds with pure standards. The concentrations and signature peaks are provided in Table 2.

### 4.5. Effect of Hypoxis Hemerocallidea Leachate on Tomato Seed Germination

Seeds of a determinate tomato (*Solanum lycopersicum* L.) variety, Floradade, were procured from the local market. Twenty-five seeds, replicated three times, were placed in each petri dish (90 mm diameter) lined with filter paper moistened with the leachate or distilled water. The seeds were exposed to *H. hemerocallidea* leachate prepared as described in Section 4.1 above, by adding 5 mL of the leachate into each petri dish. As a control, another set of seeds was treated with 5 mL of distilled water. The petri dishes were placed randomly in a growth chamber set at a constant temperature of 25 °C and in a constant state of darkness for five days. An additional 2 mL of the leachate or distilled water was added to the respective petri dishes on day four of the experiment to avoid the drying of the seeds.

Seed germination data were collected on a daily basis beginning from the first day after setting up the experiment. The final germination percentage, mean germination time, germination index, germination rate index, and coefficient of velocity of germination were calculated using the following formulae [79,80]:Final germination percentage (FGP) = Final number of germinated seeds in a seed lot × 100
Mean germination time (MGT) = ∑*f*∙x/∑*f*
where *f* = number of seeds germinated on day x.
Germination index (GI) = (5 × n1) + (4 × n2) + … + (1 × n5)
where n1, n2…n4 = number of germinated seeds on the first, second, and subsequent days until the last day of counting; 5, 4… represent the weights given to the number of germinated seeds on the first, second, and subsequent days, respectively.
Germination rate index (GRI) = [(G_1_/1) + (G_2_/2) + … + (G_t_/*t*)]
where G_1_, G_2_, and G_t_ represent the percentage of seed germination on day 1, 2, and *t*.
Coefficient of velocity of germination (CVG) = 100 × ∑Ni/∑NiTi
where Ni = number of germinated seeds per day, Ti = number of days from start of the experiment.

The data were subjected to a two-sample *t*-test using Genstat^®^ 64-bit Release 21.1 (PC/Windows 8-10, ver. 21.1, VSN, Rothamsted, UK).

## 5. Conclusions

*Hypoxis hemerocallidea* plants are generally dormant in the winter due to low soil moisture and decreasing temperatures, among other factors. The dormant state starts with leaf senescence, during which the lactose content could increase. Under natural conditions, the NADES compounds could be leached out by rainfall, resulting in increased concentrations or ratio of dormancy-breaking hormones. This correlates with exposure to distilled water through the soaking of corms, as applied in the current study. The study is the first to successfully identify four compounds that could have an inhibitory effect on the growth and development of *H. hemerocallidea* corms. Choline, lactose, propylene glycol, and succinate are reported to be part of NADES compounds that protect plant cells during stress period. The four growth-retarding compounds were leached out of the corms, possibly resulting in a balanced hormonal concentration for breaking bud dormancy and for cormlet development. The harvesting period of *H. hemerocallidea* corms could be critical in medicinal use since higher succinate concentrations could lead to kidney damage, whereas lactose could be problematic for lactose-intolerant individuals. However, there are some therapeutic benefits of the compounds that need to be further investigated. The possibility of producing an alternative to the petroleum-based propylene glycol, which may be less toxic, also warrants further investigations.

## Figures and Tables

**Figure 1 plants-11-02387-f001:**
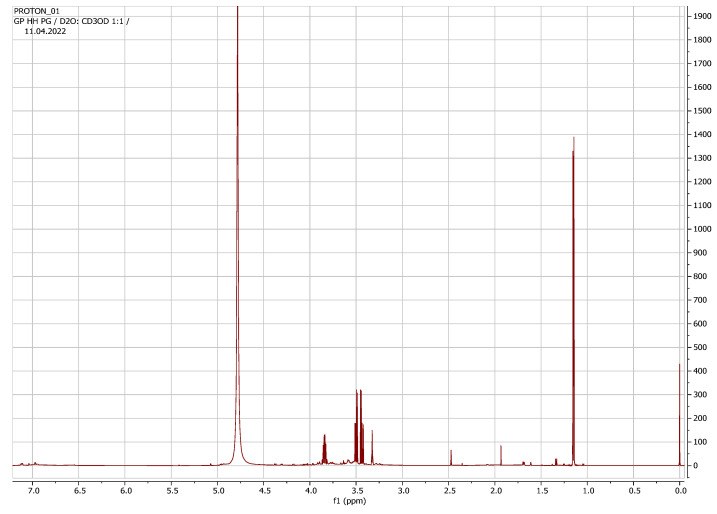
The ^1^H-NMR full spectrum of *Hypoxis hemerocallidea* leachate metabolite profile.

**Figure 2 plants-11-02387-f002:**
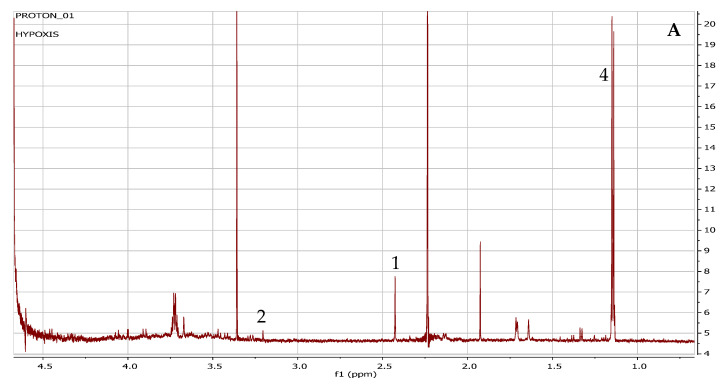
H-NMR spectra showing the metabolite profile of the leachate from *Hypoxis hemerocallidea* corms. 1 = choline and 2 = succinate (**A**), 3 = lactose and 4 = propylene glycol (**B**).

**Figure 3 plants-11-02387-f003:**
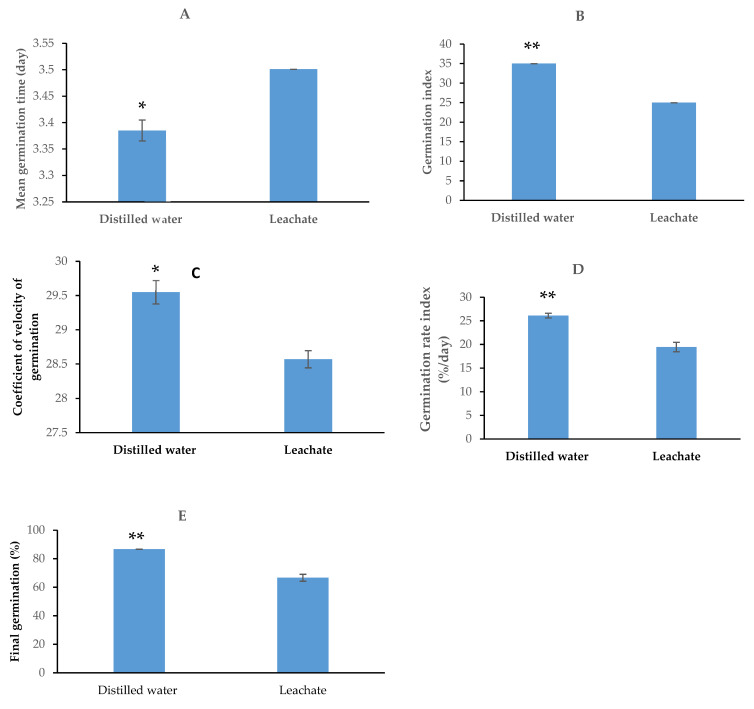
Effect of *Hypoxis hemerocallidea* leachate on tomato seed germination: mean germination time (**A**), germination index (**B**), coefficient of velocity of germination (**C**), germination rate index (**D**), and final germination percentage (**E**). *, ** = significant at *p* < 0.10 or *p* < 0.05, respectively.

**Figure 4 plants-11-02387-f004:**
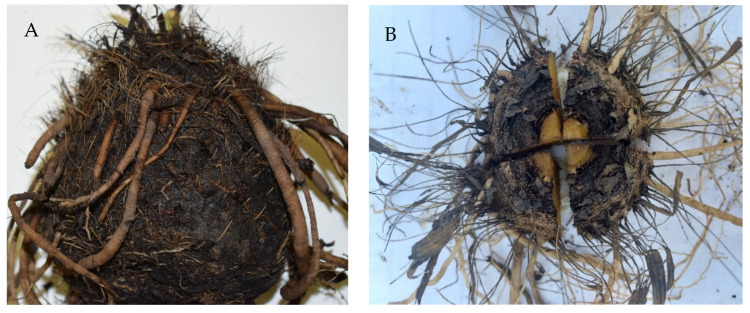
A full *Hypoxis hemerocallidea* corms (**A**) and the top view of the corm cut into four pieces (**B**).

**Table 1 plants-11-02387-t001:** Annotation of ^1^H-NMR spectral regions and chemical shifts of the four compounds identified from the *Hypoxis hemerocallidea* leachate.

Compounds	Chemical Group	^1^H-NMR Chemical Shifts (ppm)	Reference Chemical Shifts (ppm)	Chenomx (ppm)	References	Human Metabolome Database	Assigned Number
Choline (C_5_H_14_NO)	Organic acid	3.192	3.18 3.22	3.19	[27] [28]	3.18 3.50 4.05	**1**
Succinate (C_4_H_6_O_4_)	Organic acid	2.42	2.39	2.45	[29]	2.39	**2**
Lactose (C_12_H_22_O_11_)	Sugar	3.25 3.55 3.75 3.85 3.95 5.12	3.29 3.67 3.73 4.45 5.23	3.23 3.54 3.69 3.85 3.9 4.35 5.17	[27]	3.28 3.55 3.60 3.66 3.73 3.79 3.86 3.94 4.45 5.22	**3**
Propylene glycol (C_3_H_8_O_2_)	Alcohol	1.14 1.16 3.4 3.5		1.13 1.15 3.42 3.48		1.12 1.13 3.4 3.5 3.9	**4**

**Table 2 plants-11-02387-t002:** qNMR of the compounds identified with Chenomx software using TSP at 0.1% as reference compound and indicating the signature peaks used for the quantification of each compound. (s) = singlet, (d) = doublet and (t) = triplet.

Compound	Concentration (mM)	Signature Peak (ppm)
Propylene glycol	0.0038	1.13, 1.14 (d)
Lactose	0.26	3.21, 3.23, 3.25 (t)
Choline	0.0045	3.19 (s)
Succinate	0.04	2.42 (s)

## Data Availability

The data presented in this study are openly available in Mendeley database; Prinsloo, Gerhard; Mofokeng, Meshack (2022), “Hypoxis leachate”, Mendeley Data, V1, doi: 10.17632/z4wt4jpdd8.1.

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
