# Peer review of "NADES Compounds Identified in *Hypoxis hemerocallidea* Corms during Dormancy"

_plants, 2022, doi:10.3390/plants11182387_

Round 1

Reviewer 1 Report

The present paper has been carried out correctly. NMR analysis associated to the the use of standard spikes makes clear the identification of the selected compounds. The results are intriguing, indicating that the proposed cultivation system can promote the vegetative reporduction of H. hemerocallidea potentially usable as medicinal plant. However, my major concern is that the research is very limited and report just the evidence of one molecular technique. I suggest to the authors to improve their work and resubmit the paper. For instance, no data are reported about changes in plant morphology and physiology or about the content of medicinal compounds from the investigated H. hemerocallidea corms. I also suggest to convert Figure 1 as a supplemental material and insert some photos of the corms. The paper, althought well-structured for its content, does not contain enough data to reach the final conclusions reported in the text and however not to be published on the present type of journal with high publication standards. I will be happy to revise the same paper if highly improved as suggested.

-

Author Response

Reviewer #1

The present paper has been carried out correctly. NMR analysis associated to the use of standard spikes makes clear the identification of the selected compounds. The results are intriguing, indicating that the proposed cultivation system can promote the vegetative reproduction of H. hemerocallidea potentially usable as a medicinal plant.

Response – The authors are thankful to the reviewer for the positive and encouraging general comment.

Comment 1.1. However, my major concern is that the research is very limited and report just the evidence of one molecular technique. I suggest to the authors to improve their work and resubmit the paper. For instance, no data are reported about changes in plant morphology and physiology or about the content of medicinal compounds from the investigated H. hemerocallidea corms.

Response – the comment is appreciated. The paper focuses on the NMR analysis to identify potential inhibitory compounds which could affect corm bud development. This work follows previous work, which proved that soaking corms in water achieved similar results as soaking in plant growth regulators. The content of medicinal compounds have been added in the general introduction, in Line 67-77. Additionally, qNMR of the identified compounds have been included in Table 2.

Comment 1.2.  I also suggest converting Figure 1 as supplemental material and insert some photos of the corms.

Response – Figure 1 has been converted into supplementary material and a new figure (full NMR spectrum) has been added as per the suggestion by reviewer 3. The photos of the corm are included in the materials and methods. The graphical abstract also has photos of the plant and the corm.

Comment 1.3. The paper although well-structured for its content, it does not contain enough data to reach the final conclusions reported in the text and however not to be published on the present type of journal with high publication standards. I will be happy to revise the same paper if highly improved as suggested.

Response – Data on the inhibitory effect of the leachate has been added to assist in confirming the inhibitory effect, as per the suggestion by the reviewer  as well as the concentrations of the compounds in Table 2 as suggested by reviewer 5

Reviewer 2 Report

In such an article, it is desirable to review the work on inhibitors and stimulants isolated from plants. And also add an analysis of works on allelopathic interactions in the plant world.

The review of published works may include not only publications of the last 15-20 years, but the search should be deeper when there was a lot of interest in this topic. These are works of the 60s and 70s of the twentieth century. It is always interesting in the article to learn more about the medicinal properties of the studied plant species, and about the features of their use in traditional medicine. And various other uses. Electronic journals allow the presentation of literature analysis without major restrictions.

It would be good to expand the review of publications in terms of the study of water-soluble inhibitors and stimulants of compounds, primary and secondary metabolites synthesized by plants. Paying more attention to the possible role of these compounds in the life of plants and the existence of the species in the cenosis.

Author Response

Comment 2.1. In such an article, it is desirable to review the work on inhibitors and stimulants isolated from plants. And also add an analysis of works on allelopathic interactions in the plant world.

Response – Literature was added on inhibitors and stimulants. Allelopathic interactions would be an interesting addition; however, as this is not within the scope and aligns with the study's findings, it was not included. The literature on inhibitors and stimulants were added in lines 58 to 63.

Comment 2.2. The review of published works may include not only publications of the last 15-20 years, but the search should be deeper when there was a lot of interest in this topic. These are works of the 60s and 70s of the twentieth century.

Response – Thank you for the comments. Recent literature was added.

Comment 2.3. It is always interesting in the article to learn more about the medicinal properties of the studied plant species, and about the features of their use in traditional medicine. And various other uses. Electronic journals allow the presentation of literature analysis without major restrictions.

Response – The plant's medicinal properties and uses are added in lines 67 to 76.

Comment 2.4. It would be good to expand the review of publications in terms of the study of water-soluble inhibitors and stimulants of compounds, primary and secondary metabolites synthesized by plants. Paying more attention to the possible role of these compounds in the life of plants and the existence of the species in the cenosis.

Response – Thank you for the comments. The literature review was expanded, and additional information was added (lines 58-63 and 68-77).

Reviewer 3 Report

 The leaching out of compounds  from roots of  Hypoxis hemerocallidea in distilled water improved the propagation and development of cormlets, suggesting the potential leaching out of inhibitory chemical compounds. NMR spectral data of the leachate from dormant 18 H. hemerocallidea corms was investigated. On fact an improvement of the  functions of the plant was suggested  an inhibitory effect of these compounds.  

The assignments performed by literature data and confirmed by comparison  with standards.Choline and succinate, propylene glycol and lactose were identified  as inhibitory compounds.

The paper is interesting and reports an original approach to the metabolism of plants particularly in the dormancy period. Thus I suggest the publication. Just few observations that can be easily  reported to the AAs.

  The AAs reported that “After 16 h exposure of corms to distilled water, the water had changed colour from  a clear to dark brown/blackish liquid, indicative of some chemicals having leached out  from the corm pieces.” But the AAs did not say if the release in the water continues after the first observation.

I believe that a complete presentation of the NMR spectrum may better help the reader to understand  the object of the study.  

The procedure used for the spiking with the standard compounds (Figures 79 1A-G) was not descrive carefully. The figures report  the spectra for the compound identification but    the ratio with the natural concentration is not easily valuable.

It is important to know which is the purity of the standards  used. Particularly for the gallic acid. I feel  from my NMR experience that for the spectrum in the supplementary   the assignment would be performed safety. Considering  the high amplification of the spectrum used a verification of the purity of the solvents and of the standards  and , also. about the impurities  usually  adsorbed in the standard NMR tubes as well as in TSP would be useful to identify the resonance at 7.05. The non assignment I feel is an excess of honesty. Help may be the reading of  the paper of (Gregory R. Fulmer,*,† Alexander J. M. Miller,‡ Nathaniel H. Sherden,‡

Hugo E. Gottlieb,§ Abraham Nudelman,§ Brian M. Stoltz,‡ John E. Bercaw,‡ and Karen I. Goldberg†, J Organometallic chemistry, 2010, 29, 2176–2179).

It is important to comment that the retarding effet ( inhibitory) was logical derived from the leached compounds but that they have an are inhibitory effect is at the moment a suggestion. In fact an experimental proof of this sentence was not achieved adding these compounds to a normal plant.

The discussion about the metabolic effect of succinate was obtained from the literature and it is commented properly.

The complete spectrum of lactose was not reported in details particularly  for the resonance at 5.2 ppm  which can be considered diagnostic of this compound. As already told in advance a complete NMR spectrum should be shown.

Propylene glycol is an aliphatic alcohol used as a deep eutectic solvent (DES) beyond the use  as normal antifreezing in the normal use,  was also identified. On the other hand the gibberellin hormones were not identified. Thus the discussion about it should be more hypothesized

Author Response

The leaching out of compounds from roots of Hypoxis hemerocallidea in distilled water improved the propagation and development of cormlets, suggesting the potential leaching out of inhibitory chemical compounds. NMR spectral data of the leachate from dormant H. hemerocallidea corms was investigated. In fact, an improvement of the functions of the plant suggested an inhibitory effect of these compounds.  The assignments were performed by literature data and confirmed by comparison with standards. Choline and succinate, propylene glycol, and lactose were identified as inhibitory compounds. The paper is interesting and reports an original approach to the metabolism of plants, particularly in the dormancy period. Thus I suggest the publication. Just a few observations that can be easily reported to the AAs.

Response – The positive general comment is appreciated and well received.

Comment 3.1. The AAs reported that “After 16 h exposure of corms to distilled water, the water had changed colour from  a clear to dark brown/blackish liquid, indicative of some chemicals having leached out  from the corm pieces.” But the AAs did not say if the release in the water continues after the first observation.

Response – In our previous work, as indicated in the introduction (Line 84), soaking for 120 minutes improved cormlet development, suggesting that the compounds were leached out. To avoid confusion, the sentence “After 16 h exposure of corms to distilled water, the water had changed colour from a clear to dark brown/blackish liquid, indicative of some chemicals having leached out from the corm pieces.” have been removed from the results section and rephrased in the materials and methods.

Comment 3.2. I believe that a complete presentation of the NMR spectrum may better help the reader to understand  the object of the study.  

Response - A complete NMR analysis is provided (as shown in figure 1). Additionally, the NMR file shave been deposited in a Mendeley database (Prinsloo, Gerhard; Mofokeng, Meshack (2022), “Hypoxis leachate”, Mendeley Data, V1, doi: 10.17632/z4wt4jpdd8.1). This allows access to all the NMR analysis.

Comment 3.3. The procedure used for the spiking with the standard compounds (Figures 79 1A-G) was not fully described.

Response - The section was revised to provide more clarity on the method used and the changes are provided as track changes in lines 384-391.

Comment 3.4. The figures report  the spectra for the compound identification but  the ratio with the natural concentration is not easily valuable.

Response - All the analyses were deposited in the Mendeley database and is publicly available (Prinsloo, Gerhard; Mofokeng, Meshack (2022), “Hypoxis leachate”, Mendeley Data, V1, doi: 10.17632/z4wt4jpdd8.1). Quantification was also included in Table 2 using Chenomx software and TSP  as reference standard. The method was added under section 4.4 in lines 300-307.

Comment 3.5. It is important to know the purity of the standards used.

Response – The purity of the standards have been included under the new heading “4.2 Reagents” in lines 345-349.

Comment 3.6. Particularly for the gallic acid. I feel from my NMR experience that for the spectrum in the supplementary the assignment would be performed safety. Considering  the high amplification of the spectrum used a verification of the purity of the solvents and of the standards  and , also. about the impurities  usually  adsorbed in the standard NMR tubes as well as in TSP would be useful to identify the resonance at 7.05. The non assignment I feel is an excess of honesty. Help may be the reading of  the paper of (Gregory R. Fulmer,*,† Alexander J. M. Miller,‡ Nathaniel H. Sherden,‡

Hugo E. Gottlieb,§ Abraham Nudelman,§ Brian M. Stoltz,‡ John E. Bercaw,‡ and Karen I. Goldberg†, J Organometallic chemistry, 2010, 29, 2176–2179).

Response - The authors additionally used Chenomx to support the identification of the compounds. The data is now also included in Table 1. The peak for gallic acid of the pure standard at 7.05 is not aligning with the peak at 7.035 of the sample, which was also confirmed with the use of Chenomx..

Comment 3.7. It is important to comment that the retarding effect (inhibitory) was logically derived from the leached compounds but that they have an inhibitory effect is at the moment a suggestion. In fact, an experimental proof of this sentence was not achieved by adding these compounds to a normal plant.

Response - The authors acknowledges the comment and the line has been changed to “These compounds are supposedly leached out of the corms during the first rains under natural conditions, possibly accompanied by changes in the ratios of dormancy-breaking hormones and inhibitory compounds, to release bud dormancy”. The leachate was tested on seed germination of tomato seeds and results, discussions and methodology are included (lines 160-169, 283-293 and 397-437, respectively).

Comment 3.8. The discussion about the metabolic effect of succinate was obtained from the literature and it is commented properly.

Response – The comment is appreciated.

Comment 3.9. The complete spectrum of lactose was not reported in details particularly  for the resonance at 5.2 ppm  which can be considered diagnostic of this compound. As already told in advance a complete NMR spectrum should be shown.

Response - The spectrum showing the anomeric proton at 5.17 ppm has been included in the supplementary material as figure S5.

Comment 3.10. Propylene glycol is an aliphatic alcohol used as a deep eutectic solvent (DES) beyond the use  as normal antifreezing in the normal use,  was also identified. On the other hand the gibberellin hormones were not identified. Thus the discussion about it should be more hypothesized

Response – The comment is noted and the line has been changed to “These compounds are supposedly leached out of the corms during the first rains under natural conditions, possibly accompanied by changes in the ratios of dormancy-breaking hormones and inhibitory compounds, to release bud dormancy.”

Reviewer 4 Report

Dear Authors,

Your manuscript "NADES Compounds Identified in Hypoxis hemerocallidea Corm during Dormancy" is interesting.

In my opinion, the "Introduction" part is well written, it gives information about a goal you have set. Although the quoted sources are very old, and the new ones from the last 3-5 years are four, two of which are self-citations. I think the authors could update the references they cite in this section.

In the Methods and Materials section, the authors have accurately and clearly described the methods they used. In my opinion, any researcher could repeat them. I think that according to UPAC the compound used by the authors is spelled sodium salt of 3-(trimethylsilyl)propionic acid and I think it is more correct to spell it that way. Also, (CH3OH-d4) would be better written as CD3OD.

The authors write the following: "....soil particles were removed by rinsing in chlorine-free tap water..." after the tap water, i.e. it is drinking water and this water is used by the population. How then is the water used by the population purified, after the authors write that there is no chlorine, perhaps it is ozonated or other preparations are used? Perhaps the authors mean that their university has a distilled water tap that they used.

In my opinion, to prove the relevant compounds such as choline, succinate, lactose, propylene glycol in the plant Hypoxis hemerocallidea, in addition to 1H-NMR analysis, the authors should also do IR, I think it would be better FTIR, mass spectrometry. elemental analysis, since NMR alone does not give a complete picture are the results.

Results are presented accurately and clearly. The discussion corresponds to the presented results and is supported by relevant citations 16 from the last 3-5 years and one self-citation.

There is a typo in References 46 to 59, I give the following example of what I mean: 48. [48] Harvanko A, Kryscio R, Martin C, Kelly T. 2019. Stimulant effects of propylene glycol and vegetable glycerin in e-cigarette liquids. Alcohol addiction to drugs. 194: 326-329.

The conclusion supports the results obtained by the authors.

Author Response

Dear Authors,

Your manuscript "NADES Compounds Identified in Hypoxis hemerocallidea Corm during Dormancy" is interesting.

Response – The authors are grateful that the manuscript was found to be interesting by the reviewer.

Comment 4.1. In my opinion, the "Introduction" part is well written, it gives information about a goal you have set. Although the quoted sources are very old, and the new ones from the last 3-5 years are four, two of which are self-citations. I think the authors could update the references they cite in this section.

Response – The cited references have been updated, taking into consideration the relevance of all the cited references (old and new) and the comment (2.2) by reviewer #2.

Comment 4.2. In the Methods and Materials section, the authors have accurately and clearly described the methods they used. In my opinion, any researcher could repeat them. I think that according to UPAC the compound used by the authors is spelled sodium salt of 3-(trimethylsilyl)propionic acid and I think it is more correct to spell it that way. Also, (CH3OH-d4) would be better written as CD3OD.

Response – the positive comment is acknowledged. The correction was made as suggested in lines 353, 284 and 368.

Comment 4.3. The authors write the following: "...soil particles were removed by rinsing in chlorine-free tap water..." after the tap water, i.e. it is drinking water and this water is used by the population. How then is the water used by the population purified, after the authors write that there is no chlorine, perhaps it is ozonated or other preparations are used? Perhaps the authors mean that their university has a distilled water tap that they used.

Response – The comment is acknowledged and to avoid any potential confusion, chlorine-free tap water was changed to distilled water.

Comment 4.4. In my opinion, to prove the relevant compounds such as choline, succinate, lactose, propylene glycol in the plant Hypoxis hemerocallidea, in addition to 1H-NMR analysis, the authors should also do IR, I think it would be better FTIR, mass spectrometry. elemental analysis, since NMR alone does not give a complete picture are the results.

Response - The authors additionally report the use of Chenomx, which is a database for matching potential candidates in a sample (line 384). Also, the use of spiking of the sample with authentic standards to confirm the annotation of the compounds is identification at level 1 (Sumner et al., 2007), which has been included in line 391.

Comment 4.5. Results are presented accurately and clearly. The discussion corresponds to the presented results and is supported by relevant citations 16 from the last 3-5 years and one self-citation.

Response – The comment is appreciated and acknowledged.

There is a typo in References 46 to 59, I give the following example of what I mean: 48. [48] Harvanko A, Kryscio R, Martin C, Kelly T. 2019. Stimulant effects of propylene glycol and vegetable glycerin in e-cigarette liquids. Alcohol addiction to drugs. 194: 326-329.

Response – The typo in the reference list, from references #58 – 72, has been corrected.

The conclusion supports the results obtained by the authors.

Response – The authors acknowledge the positive comment. 

Reviewer 5 Report

This paper describes the NADES compounds identified in Hypoxis hemerocallidea corm during dormancy. The article is quite complete, it is of interest to the scientific community, the methods and statistics used are appropriate and the results are conveniently described. The work is well discussed and is supported by the references provided by the authors. The work is interesting and delves into the compounds present in plants for dormancy. On the other hand, I consider that the Materials and Methods” section should be described in greater depth and scientific rigor.

I consider that the article is appropriate to be published in Plants journal once the authors have made the following modifications to it.

The authors must perform the quantification of the compounds present in the samples. They are simple compounds to quantify.

“Materials and Methods” section: This section should be described in greater depth and scientific rigor.

-          Include a "Plant material" subsection. This section must describe where the plants were obtained from, the number of plants collected, how the species in question is guaranteed, etc. When plants were cut, specify size of cuts, weight of pieces, etc. The authors must describe this part with scientific rigor.

-          Include a "Reagents" section. In this section describe the reagents used, solvents, commercial brands, etc.

-          Include the city and country of all the companies cited, and cite the companies of all the reagents and equipment’s employed. In case of USA companies, include the city and the state abbreviation. Unify and apply to the entire document.

References: Use the format of the journal.

Minor considerations:

Line 103: “align.”.

Line 106: “Hypoxis hemerocallidea” in italics.

Line 110: [6,8,18]

Lines 166, 168, 177,..: Put “β” in italics.

Line 227: Include “g” or the orbital radius.

Lines 221-223, 235, etc.: “mL” instead of “ml”. Unify and apply to the entire document.

Lines 225, ….: Do not capitalize “Trimethylsilylpropionic”. Unify and apply to the entire document.

Author Response

This paper describes the NADES compounds identified in Hypoxis hemerocallidea corm during dormancy. The article is quite complete, it is of interest to the scientific community, the methods and statistics used are appropriate and the results are conveniently described. The work is well discussed and is supported by the references provided by the authors. The work is interesting and delves into the compounds present in plants for dormancy. On the other hand, I consider that the Materials and Methods” section should be described in greater depth and scientific rigor. I consider that the article is appropriate to be published in Plants journal once the authors have made the following modifications to it.

Response – The positive comment and consideration by the reviewer is appreciated by the authors.  

Comment 5.1. The authors must perform the quantification of the compounds present in the samples. They are simple compounds to quantify.

Response - The quantification has been done using Chenomx software and the TSP peak at 0.1% as reference. The concentrations are provided in Table 2 and the methodology is added under section 4.4 in lines 370-381.

Comment 5.2. “Materials and Methods” section: This section should be described in greater depth and scientific rigor.

Include a "Plant material" subsection. This section must describe where the plants were obtained from, the number of plants collected, how the species in question is guaranteed, etc. When plants were cut, specify size of cuts, weight of pieces, etc. The authors must describe this part with scientific rigor.

Response – The “Plant material” subsection has been added as per the recommendation.

 Include a "Reagents" section. In this section describe the reagents used, solvents, commercial brands, etc.  Include the city and country of all the companies cited, and cite the companies of all the reagents and equipment’s employed. In case of USA companies, include the city and the state abbreviation. Unify and apply to the entire document.

Response – The “Reagents” subsection has been added following the recommendation.

Comment 5.3. References: Use the format of the journal.

Response – The references have been formatted accordingly.

Round 2

Reviewer 1 Report

In this new version the paper is more adequate to this journal. I have just a minor comment for the authors: at line 496, they talk about ROS increase during plant growth, development, and stress response. However, they support this citation with just one reference that does not focuses specifically on plant stress mechanisms. I suggest to add other references here, such as: Journal of plant research, 2019, 132.3: 439-455; Plant, Cell & Environment, 2012, 35.2: 281-295; Antioxidants, 2021, 10.07: 1048; Plant Physiology and Biochemistry, 2019, 141: 353-369; Agronomy, 2020, 10.7: 1014. After that, the paper can be published.

Author Response

The authors appreciate the reviewer's positive and encouraging general comment. Lines 233 have been updated to include the suggested references, and the references list has been updated accordingly.

Reviewer 4 Report

Dear Authors,

I agree with the corrections you have made to the manuscript.

Author Response

The authors are thankful to the reviewer for the positive and encouraging general comment.

Reviewer 5 Report

The authors have made the indicated modifications and the article has improved substantially. For this reason, I consider that the article can be considered for publication in Plants journal in its current form.

Author Response

(The authors gave the same response as above.)
